# Inhibition of BRD4 Reduces Neutrophil Activation and Adhesion to the Vascular Endothelium Following Ischemia Reperfusion Injury

**DOI:** 10.3390/ijms21249620

**Published:** 2020-12-17

**Authors:** Shelby Reid, Noah Fine, Vikrant K. Bhosle, Joyce Zhou, Rohan John, Michael Glogauer, Lisa A. Robinson, James W. Scholey

**Affiliations:** 1Institute of Medical Sciences, University of Toronto, Toronto, ON M5S 1A8, Canada; xiaohua.zhou@utoronto.ca (J.Z.); lisa.robinson@sickkids.ca (L.A.R.); james.scholey@utoronto.ca (J.W.S.); 2Matrix Dynamics Group, Faculty of Dentistry, University of Toronto, Toronto, ON M5G 1G6, Canada; nfine1@gmail.com (N.F.); Michael.Glogauer@dentistry.utoronto.ca (M.G.); 3Program in Cell Biology, The Hospital for Sick Children, Peter Gilgan Centre for Research and Learning, Toronto, ON M5G 0A4, Canada; vikrant.bhosle@mail.mcgill.ca; 4Department of Laboratory Medicine and Pathobiology, University Health Network, Toronto, ON M5G 2C4, Canada; rohan.john@uhn.ca; 5Department of Dental Oncology, Maxillofacial and Ocular Prosthetics, Princess Margaret Cancer Centre, Toronto, ON M5G 2M9, Canada; 6Centre for Advanced Dental Research and Care, Mount Sinai Hospital, Toronto, ON M5G 1X5, Canada; 7Department of Pediatrics, Faculty of Medicine, University of Toronto, Toronto, ON M5G 1X8, Canada; 8Division of Nephrology, The Hospital for Sick Children, Toronto, ON M5G 1X8, Canada; 9Department of Medicine, Division of Nephrology, University Health Network, Toronto, ON M5G 1X8, Canada

**Keywords:** ischemia-reperfusion, NFκB, BRD4, inflammation, neutrophils, tubule injury

## Abstract

Renal ischemia reperfusion injury (IRI) is associated with inflammation, including neutrophil infiltration that exacerbates the initial ischemic insult. The molecular pathways involved are poorly characterized and there is currently no treatment. We performed an in silico analysis demonstrating changes in NFκB-mediated gene expression in early renal IRI. We then evaluated NFκB-blockade with a BRD4 inhibitor on neutrophil adhesion to endothelial cells in vitro, and tested BRD4 inhibition in an in vivo IRI model. BRD4 inhibition attenuated neutrophil adhesion to activated endothelial cells. In vivo, IRI led to increased expression of cytokines and adhesion molecules at 6 h post-IRI with sustained up-regulated expression to 48 h post-IRI. These effects were attenuated, in part, with BRD4 inhibition. Absolute neutrophil counts increased significantly in the bone marrow, blood, and kidney 24 h post-IRI. Activated neutrophils increased in the blood and kidney at 6 h post-IRI and remained elevated in the kidney until 48 h post-IRI. BRD4 inhibition reduced both total and activated neutrophil counts in the kidney. IRI-induced tubular injury correlated with neutrophil accumulation and was reduced by BRD4 inhibition. In summary, BRD4 inhibition has important systemic and renal effects on neutrophils, and these effects are associated with reduced renal injury.

## 1. Introduction

Acute kidney injury (AKI) is responsible for approximately 5% of all hospitalizations, is associated with increased morbidity and mortality, as well as progression to chronic kidney disease (CKD) and end-stage renal disease (ESRD) [1]. Nearly two-thirds of all AKI cases are secondary to renal ischemia-reperfusion injury (IRI) [2]. Despite recent advances, a better understanding of the pathogenesis of IRI is needed to identify new targets for therapy.

The early phase of IRI (up to 48 h post-IRI) is associated with inflammation, particularly, recruitment of circulating neutrophils to the kidney [3,4]. Neutrophils adhere to the endothelium and transmigrate to the site of injury through the synergy of numerous adhesion molecules [5,6]. Besides clogging peritubular capillaries, neutrophils cause tissue injury by releasing proteases and pro-inflammatory cytokines, and generating reactive oxygen species (ROS) [7,8,9].

Nuclear factor-κB (NFκB)-mediated gene expression is a hallmark of tissue inflammation [10]. The NFκB signalling pathway is initiated through activation of inhibitor of NFκB (IκB) kinases, which in turn, results in phosphorylation and ubiquitin dependent degradation of IκB kinase β (IκBβ) [11]. This enables NFκB, a heterodimer of NFκB subunit 1 (p50) and RELA proto-oncogene, NFκB subunit (RelA), to be released, with subsequent translocation into the nucleus to regulate transcription of pro-inflammatory genes [12].

The effect of NFκB on gene expression is further altered by post translational modification, including acetylation [13]. Specifically, acetylation of RelA at lysine-310 enables bromodomain-containing protein 4 (BRD4), a bromodomain and extra-terminal (BET) protein, to bind to NFκB [14]. This protein-protein interaction enhances the transcriptional activity of NFκB [14]. The important role of BRD4 in NFκB-mediated gene expression has led to the development of small molecule inhibitors that competitively bind to acetylated RelA [15,16]. While several BET inhibitors have been developed, a small molecule named MS417 has shown great promise in attenuating injury in different models of kidney injury. MS417 was designed to specifically inhibit BRD4 from interacting and binding to the lysine-310 residue on the acetylated RelA, effectively inhibiting transcription from occurring [16]. MS417 has been shown to limit chronic kidney injury in murine models of HIV nephropathy and diabetic nephropathy through reduced expression of NFκB target genes and improved renal function [16,17].

Accordingly we sought to study the effect of BRD4 inhibition on the early phase of inflammation that characterizes renal IRI. We first performed an in silico analysis of NFκB-mediated gene expression in early IRI, Next, we examined the effect of BRD4 inhibition on neutrophil adhesion to cultured endothelial cells and NFκB-mediated gene expression in kidney tubule cells. Subsequently, we studied the effect of BRD4 inhibition in a murine model of IRI. In vitro, we observed that BRD4 inhibition reduces neutrophil adhesion to the endothelium following hypoxia re-oxygenation and decreases inflammatory gene expression in primary tubule cells. In vivo, BRD4 treatment attenuates NFκB-mediated gene expression, reduces neutrophil recruitment and activation, and decreases tubular injury.

## 2. Results

### 2.1. NFκB-mediated Gene Expression Characterizes Early IRI

The activation of canonical NFκB signaling is one of the primary mechanisms regulating gene expression in the setting of tissue inflammation, and has been implicated in the renal response to changes in the redox state that accompanies IRI [18]. To assess changes in NFκB-mediated gene expression in the renal IRI setting, we performed an in silico analysis of changes in early gene expression using publicly available data from Liu et al. [19]. We first assembled a list of 113 genes transcriptionally regulated by NFκB based on work by Pahl et al. [20]. Thirty-four of these genes were identified in the dataset compiled by Liu et al. (Figure 1A) [19,20]. Significantly enriched biological processes of the 34 NFκB-mediated genes included surface receptor signaling, transcription regulator activity, monocyte chemotaxis and leukocyte activation (Figure 1B). Euclidean hierarchical clustering showed differential regulation of NFκB-mediated genes at 4, 24 and 48 h of IRI (Figure 1C). Following Benjamini and Hochberg corrections for multiple testing, 6, 11 and 12 genes remained differentially expressed at 4, 24 and 48 h, respectively (Figure 1D–G, Appendix A).

### 2.2. Inhibiting NFκB-Mediated Gene Expression, In Vitro

Canonical NFκB signaling pathway is regulated by the BET protein BRD4 [14]. We chose to target NFκB signaling with the BET-specific inhibitor, MS417, which was designed to inhibit BRD4 from interacting and binding to acetylated NFκB [16]. To confirm the intended inhibitory effect of the BRD4 inhibitor, MS417, we performed a NFκB luciferase activity assay (Figure 2A) [16]. Tumor necrosis factor (TNF)-α was used as a positive control based on its known activation of NFκB [21]. Since the S3 segment of the proximal tubule is a major target in renal IRI, we used the proximal tubule epithelial cell line, human kidney-2 (HK-2) cells [22]. When HK-2 cells were pre-treated with 1 μM MS417, NFκB-mediated luciferase activity was reduced to near basal levels.

Recent studies suggested that BRD4 inhibition can also affect cell proliferation through c-Fos/activator protein 1 (AP-1) signaling and fibrosis through transforming growth factor (TGF)-β signaling [23,24,25]. To test this, we performed luciferase assays targeting AP-1 and TGF-β/SMAD activity due to their key roles in inflammation following IRI [26,27] (Figure 2B,C). The early phase of inflammation is followed by a regenerative phase characterized by cell proliferation, and the latter phase of IRI is characterized by fibrosis [28,29]. BRD4 inhibition with MS417 significantly reduced both AP-1 and SBE-mediated luciferase activities confirming the broader effects of BRD4 inhibition.

### 2.3. BRD4 Inhibition Blocks Neutrophil Adhesion to the Activated Endothelial Cells In Vitro

We initially performed cell viability assays to confirm MS417 has no cytotoxic effects on either endothelial cells or neutrophils. Isolated neutrophils were treated with varying concentrations of MS417 ranging from 1 nM to 1 mM (Figure 3A). Neutrophil viability was stable up to 1 μM of MS417 treatment. Human umbilical vein endothelial cells (HUVEC) treated with MS417 following hypoxia exhibited significantly improved cell viability at 6, 12 and 24 h of reoxygenation (Figure 3B). Together, these results show that MS417 has no effects on cell viability at concentrations up to 1 μM. This concentration was used for subsequent experiments.

Neutrophil recruitment is one of the first steps in the innate immune response to tissue injury. To transmigrate to the site of injury, neutrophils must first be recruited at the inner surface of the vascular endothelium [30]. Therefore, we examined if BRD4 inhibition had an effect on neutrophil adhesion to vascular endothelial cells in vitro. Neutrophil-endothelial adhesion assays were performed as previously described [31]. Combined treatment of primary HUVEC and freshly isolated human neutrophils with MS417 resulted in a significant reduction in neutrophil adhesion to TNFα-stimulated endothelial cells in static conditions (Figure 4A) [32]. These studies show that concurrent treatment of neutrophils and endothelial cells attenuate TNFα-induced adhesion.

To determine whether MS417 predominately affects neutrophils or endothelial cells, we studied both neutrophils and endothelial cells pre-treated with MS417. Pre-treatment of neutrophils alone with 1 μM MS417 prior to co-incubation with TNFα-stimulated HUVEC was sufficient to attenuate adhesion to baseline levels (Figure 4A; N + MS417). When endothelial cells were pre-treated with MS417 and stimulated with TNFα, adhesion to neutrophils was also significantly reduced (Figure 4A; H + MS417). Pre-treatment of neutrophils with MS417 had a greater effect than pre-treatment of endothelial cells (Figure 4A; N + MS417 vs. H + MS417, *p* < 0.001). Taken together, these studies show that both neutrophils and endothelial cells were affected by MS417 treatment. To better represent IRI in an in vitro setting, HUVEC were subjected to hypoxia-re-oxygenation. Following two hours of hypoxia, HUVEC were co-incubated with neutrophils treated with either 100 nM or 1 μM of MS417 and re-oxygenated for three hours. Neutrophil adhesion to endothelial cells was attenuated when neutrophils were treated with MS417 in a dose-dependent manner (Figure 4B).

### 2.4. BRD4 Inhibition Attenuates IL6 Gene Expression Following H_2_O_2_-induced Oxidative Stress

We next examined the effect of MS417 on H_2_O_2_-induced cytokine gene expression to test the effect on oxidative stress through reactive oxygen species, which has been implicated in IRI injury, in vivo [33,34]. For this, we used primary human renal epithelial cells (PTECs) as opposed to the immortalized HK-2 cells to more accurately represent the morphology of the kidney. H_2_O_2_ increased gene expression of IL6, CXCL2 and CCL2 after six hours (Figure 5A–C). MS417 pre-treatment significantly reduced H_2_O_2_-induced IL6 expression, while some reduction, although not significant, was observed for CXCL2 (Figure 5D–F).

### 2.5. BRD4 Inhibition Attenuates In Vivo CCL2 Gene Expression Following IRI

To better understand the effects of BRD4 inhibition on NFκB-mediated gene expression, an in vivo time course experiment was performed. Mice were treated with MS417 or saline (control) by oral gavage daily for seven days prior to unilateral IRI (Figure 6A). Expression of transcripts for pro-inflammatory cytokines (CCL2 and TNFα) and adhesion molecules (ICAM1 and VCAM1) was analyzed in kidney tissue by qPCR at 6, 24 and 48 h after IRI (Figure 6B–D). Expression for all four transcripts increased at all time points, post-IRI. MS417 treatment significantly reduced expression of CCL2 at 24 h, while some reduction, although not significant, was observed for ICAM1 (*p* = 0.1027).

### 2.6. BRD4 Inhibition Reduces Absolute Neutrophil Counts Following IRI

To test the effects of BRD4 inhibition on the innate immune response, in vivo, mice were subjected to IRI with or without MS417 pre-treatment. Neutrophil counts were assessed in the bone marrow, blood, and kidney during the acute and resolution phase of the inflammatory response by flow cytometry. Neutrophils were gated as Ly6G^+ve^/F4/80^−ve^ (Figure 7A). Absolute neutrophil counts in the bone marrow (Figure 7B) and blood (Figure 7C) increased significantly in sham mice 6 h after surgery compared to naïve mice, indicating that the sham surgery alone was sufficient to induce neutrophil numbers in the bone marrow and release into the circulation. Absolute counts increased in kidney tissue at 6 h-post IRI when compared to sham mice (Figure 7D) confirming recruitment to the site of injury. By 24 h-post IRI, there was a significant increase in absolute counts in the bone marrow, circulation and kidney tissue compared to sham mice. Absolute neutrophil counts remained elevated in kidney tissue with a similar trend noted in the circulation until 48 h-post IRI, while absolute neutrophil counts in the bone marrow declined (Figure 7B). MS417 treatment significantly reduced absolute neutrophil counts in the bone marrow and kidney tissue at 24 h.

Immunohistochemistry was performed to confirm the effects of BRD4 inhibition on neutrophil recruitment to the kidney in IRI. Semi-quantitative scoring of neutrophils in immuno-stained kidney sections showed a significant increase in neutrophils at 6, 24 and 48 h-post IRI compared to sham mice (Figure 8A–D). The time course of neutrophil recruitment paralleled the increase in counts observed by flow cytometry. As expected, neutrophil infiltration following IRI was predominantly located in the outer medullary zone of the kidney. MS417 treatment reduced kidney neutrophil scores at 24 h in most mice, although the overall difference between groups was not statistically significant (Figure 8A–D).

### 2.7. BRD4 Inhibition Reduces Neutrophil Up-regulation of CD66a Following IRI

Neutrophil adhesion surface markers were gated for and assessed by FACS analysis (Figure 9A,B, Appendix A). CD66a, a protein present on neutrophils that is involved in neutrophil recruitment, activation and adhesion, was significantly increased in the bone marrow at 24 h-post IRI and remained elevated until 48 h when compared to naïve mice (Figure 9C) [35,36,37]. While no change was noted in the blood (Figure 9D), CD66a expression on kidney neutrophils was elevated as early as 6 h post-IRI and remained elevated through to 48 h-post IRI (Figure 9E). Similar to absolute neutrophil counts, MS417 treatment reduced CD66a expression on neutrophils in bone marrow and kidney tissue 24 h after IRI (Figure 9C,E).

CD11b is a component on Mac-1, one of two primary integrins neutrophils express to allow binding and transmigration across the endothelium [5]. We did not observe any changes in CD11b expression on neutrophils in the bone marrow or the circulation following IRI (Figure 9F,G). However, CD11b expression on neutrophils was significantly higher in kidney tissue at 6 and 24 h post-IRI (Figure 9H). MS417 treatment had no effect on CD11b expression by neutrophils in the bone marrow, blood, or kidney tissue following IRI.

We also studied neutrophils expressing the CD markers CD55, CD101, CD62L, and CD5 due to their roles in neutrophil activation [38,39,40] (Appendix A). CD55, a complement regulatory protein which has previously been shown to be protective against renal IRI, significantly increased in kidney tissue 6 h post-IRI when compared to naïve mice (Appendix A) [41]. CD55 remained elevated until 24 h with resolution noted by 48 h post-IRI. CD101, a marker related to neutrophil maturity, was elevated in the bone marrow, blood and kidney tissue 48 h post-IRI when compared to naïve mice (Appendix A) [40]. MS417 treatment had no effect on CD55, CD101, CD62L or CD5 expression on neutrophils in the bone marrow, blood, or kidney tissue following IRI. Together, these results show there is a significant increase in absolute and activated neutrophils following IRI in the bone marrow, blood, and kidney tissue and BRD4 inhibition with MS417 reduces these counts, primarily at 24 h-post IRI.

### 2.8. BRD4 Inhibition Attenuates Tubular Injury Following IRI

Finally, we examined whether BRD4 inhibition with MS417 treatment was able to reduce tubular injury following IRI. PAS-stained sections taken at 6, 24 and 48 h post-IRI were scored in a blinded manner for standard morphologic changes of tubular damage (Figure 10A–D). As expected, there was a significant increase in tubular injury after IRI seen as early as 6 h, with further increase at 24 and 48 h. MS417 treatment significantly reduced the amount of tubular damage at 24 h post-IRI, while persistent reduction, although not significant, was also seen at 48 h (*p* = 0.0597).

We also assessed kidney injury by measuring gene expression of two classic markers of tubular injury, namely HAVCR1 (also known as KIM-1) and LCN2 (also known as NGAL) (Figure 11A–C) [42,43]. In keeping with observed histologic changes, mRNA levels for both injury markers were significantly increased at 6, 24 and 48 h after IRI. MS417 treatment reduced expression of HAVCR1 (*p* < 0.01) and LCN2 (*p* = 0.0516) 24 h after IRI.

### 2.9. Neutrophil Infiltration Correlates with Tubule Injury

Given that BRD4 inhibition with MS417 reduced neutrophil counts and tubule injury, especially at 24 h, we related kidney injury to the semi-quantitative scoring of the neutrophil accumulation, based on immunohistochemistry 24 h after IRI. There was a significant correlation between the neutrophil staining and tubular injury in our murine model of IRI (R = 0.7663, *p* = 0.004) (Figure 12). Collectively, our results show that BRD4 inhibition with MS417 ameliorates neutrophil adhesion in vitro, and reduces neutrophil recruitment, neutrophil activation, and decreases tubular injury, in vivo.

## 3. Discussion

AKI is associated with increased hospital morbidity and mortality and leads to an increased risk of developing chronic kidney disease (CKD) end-stage renal disease (ESRD) [1]. IRI is the most common cause of in-hospital AKI [2] and is characterized by an early phase of inflammation, particularly neutrophil accumulation that is associated with the release of reactive oxygen species (ROS) and pro-inflammatory cytokine production [3,4,44]. The mechanism(s) responsible for regulating this tissue response have not been fully elucidated and few treatment approaches were identified [31,45,46].

The first goal of the present study was to better understand the cellular response during the early phase of IRI-induced inflammation, of which NFκB is a major regulator. Previous studies demonstrating NFκB in the tissue response to IRI have focused on NFκB target genes and/or direct interference with NFκB activity [45,47,48]. We therefore took a systems biology approach to analyze a gene set shown to be regulated by NFκB to determine if these genes were differentially expressed following IRI. We identified 34 NFκB-mediated genes expressed in the kidney following IRI based on published microarray analyses and then studied the time-dependent effect of IRI on this gene set [19]. Unsupervised hierarchical cluster analysis showed that the expression patterns of these genes vary in early IRI. These findings show that changes in NFκB-mediated gene expression play an important role in cellular responses in early IRI, including chemotaxis, cell migration, metabolism and cell proliferation. Although other studies have shown that the canonical NFκB pathway is activated following IRI [49], the effect of NFκB blockade on IRI-induced kidney injury is incompletely understood [45,46,50]. Accordingly, our second goal was to look at the effect of NFκB blockade in renal IRI.

Our approach to block NFκB signaling focused on BET proteins. The BET protein family consists of BRD2, BRD3, BRD4 and BRDT, all of which have two conserved bromodomains at the N-terminal and an extra-terminal recruitment domain at the C-terminal [51,52]. Bromodomains regulate gene transcription by binding to acetylated residues on histones and other nuclear proteins [51]. As a NFκB coactivator, the BET protein BRD4 plays a key role in NFκB transcription [14]. BRD4 binds to the acetylated lysine-310 of RelA leading to the recruitment of the CDK9 component of the positive transcription elongation factor b (P-TEFb) complex, subsequently leading to phosphorylation of RNA polymerase II and transcriptional activation of NFκB target genes [14].

In cancer cells, the BRD4/RelA complex avoids degradation, resulting in continuous activation of NFκB, which promotes cell proliferation [53]. Thus, several small molecule inhibitors that target BRD4 have been developed [15,16]. These inhibitors have recently been explored in inflammatory disease settings given the importance of NFκB in inflammation. For example, one BRD4 inhibitor originally designed for cancer treatment, JQ1, was shown to be beneficial in mouse models of ischemic stroke, myocardial infarction, LPS-induced lung inflammation, and cisplatin-induced nephrotoxicity [34,54,55,56,57]. More recently, BRD4 inhibition was shown to target FoxO4-mediated oxidative stress following renal IRI, and reduced apoptosis and endoplasmic reticulum stress [34]. Our study extended the understanding of BRD4 inhibition with a particular focus on neutrophil infiltration following renal IRI. Taken together, these studies show that tissue response to early inflammation is highly dependent on NFκB and thus, BRD4. Furthermore, these studies suggest there is a class effect of BRD4 inhibitors and that the effects demonstrated in studies with JQ1 are not due to an off-target effect. However, the complete role of BRD4 in renal IRI has not been fully elucidated and remains a gap in our knowledge.

We first confirmed the inhibitory effect of MS417 on NFκB-mediated luciferase activity. As expected, MS417 reduced NFκB-mediated luciferase in response to TNFα. Recent studies have suggested a broader role of BRD4 outside of NFκB signaling, including effects on c-Fos/AP-1-mediated gene expression and TGF-β-mediated gene expression [23,24,25]. Given these observations, we also studied the effect of pre-treatment with MS417 on EGF-induced AP-1 luciferase activity and TGF-β-induced SBE luciferase activity. Interestingly, pre-treatment with MS417 inhibited luciferase activity in response to both of these ligands.

Taken together, these finding suggest that MS417 may have broader effects on IRI beyond the inhibition of NFκB. While we characterized the effects of BRD4 inhibition on the initial inflammatory phase, we did not assess the effects on the subsequent regeneration/repair and fibrotic phases. The regenerative phase is necessary to repair the damaged epithelium through proliferation that is primarily mediated by growth factors [58]. It is tempting to speculate that MS417 may have an impact on the cellular proliferation which characterizes the regenerative phase of IRI. Furthermore, IRI is associated with important long-term outcomes including CKD and ESRD and this phase of the natural history of IRI is characterized with interstitial fibrosis and progressive loss of function mediated, in part, by TGF-β and AP-1 mediated gene expression [29,59,60,61]. Therefore, we could speculate that MS417 treatment during this phase of IRI may be beneficial. Future studies will have to address the impact of MS417 on these different phases of IRI and the overall long-term outcomes.

We hypothesized that BRD4 inhibition would reduce neutrophil adhesion to endothelial cells. The up-regulation of endothelial adhesion molecules, including vascular cell adhesion molecule 1 (VCAM1) and intracellular adhesion molecule 1 (ICAM1), is crucial for leukocyte adhesion following IRI [62,63]. TNFα is expressed in the kidney following IRI, and is a potent activator of endothelial cell expression of these adhesion molecules [32,64,65]. MS417 treatment to both endothelial cells and neutrophils alone led to reduced neutrophil adhesion to the TNFα-activated endothelium suggesting not only reduced expression of endothelial adhesion molecules, but also a reduction in neutrophil adhesion markers. We subsequently modeled IRI by subjecting vascular endothelial cells to hypoxia re-oxygenation, in vitro, and observed a similar attenuation of neutrophil adhesion to endothelial cells with BRD4 inhibition. Our observations confirm studies performed in a model of lung inflammation where JQ1 was also shown to limit neutrophil adhesion [55]; however, treatment to neutrophils has not been previously shown. Furthermore, we extended the understanding of neutrophil biology by looking at neutrophil accumulation and activation state as defined by FACS analysis with cell surface markers, in particular CD66a and CD11b. These findings suggest that BRD4 inhibition may be an approach to limiting neutrophil accumulation in IRI-induced tissue injury, in vivo.

While we did not address the mechanisms responsible for the in vitro effect of BRD4 inhibition on neutrophil adhesion, it was previously shown that anti-CD66 monoclonal antibody treatment can inhibit neutrophil adhesion to endothelial cells [37]. Also, blockade of Mac-1 with a monoclonal antibody reduces leukocyte recruitment and adhesion [66]. Lymphocyte function-associated antigen (LFA-1; CD11a/CD18) and Mac-1 (CD11b/CD18) ligands play a cooperative role in leukocyte adhesion, while LFA-1 or Mac-1 null mice show higher rolling velocities compared to wild-type mice [5,67,68]. We saw that treatment of neutrophils alone was sufficient to limit adhesion to endothelial cells to baseline levels. Mac-1-mediated neutrophil adhesion is primarily due to binding to ICAM-1 and it has been suggested that LFA-1 may bind to other endothelial ligands [68]. BRD4 inhibition may therefore prevent expression of β2-integrins in neutrophils, an effect that could account for our observations. However, future studies are needed to better understand the mechanism underlying this effect.

We next studied the effect of BRD4 inhibition on the early accumulation of neutrophils in murine kidneys following IRI as well as on bone marrow and blood neutrophils. While we did not measure BRD4 expression following IRI, previous investigators have shown as increase in BRD4 expression following IRI and a reduction with JQ1 treatment [34]. Neutrophil production in the bone marrow in response to localized inflammation increases within minutes by as much as 10-fold [69]. These neutrophils then follow a chemotactic gradient from the bone marrow, into the circulation, and to the site of injury [70,71]. We looked at absolute neutrophil counts at 6, 24, and 48 h after IRI by FACS analysis. Our next major observation was that the sterile inflammation induced by IRI in the kidney is associated with an early increase in neutrophil numbers in the bone marrow (24 h), blood (24 h) and kidneys (6 h-48 h). Microscopic examination of kidney tissue confirmed the increase in kidney neutrophils after IRI seen by FACS analysis, and localized much of the accumulation to the outer medulla, the expected site of maximum tubular injury. These findings parallel observations in other organs. Recruitment of neutrophils to the liver following sterile injury has been shown to begin as early as 8 h with maximum recruitment at 12 h, albeit with a marked reduction by 24 h and complete resolution by 48 h [72]. The more sustained increase in kidney neutrophils that we observed could be due to the type and severity of injury.

Several studies have looked at inhibiting neutrophil accumulation in tissue by targeting different steps in recruitment including chemotaxis, activation, and transmigration [73,74,75]. In models of respiratory infection, vascular inflammation, and ischemic stroke, BRD4 inhibition reduced neutrophil infiltration at the site of injury as measured by immunohistochemical analysis [55,57,76]. Through FACS analysis, we showed that BRD4 inhibition with MS417 significantly reduced absolute neutrophil counts in the bone marrow and in the kidney tissue following IRI suggesting a decrease in neutrophil production and a decrease in infiltrating neutrophils to the site of injury. We could speculate that this is due to the down-regulation of key cytokines and chemokines mediated by NFκB that aid in the neutrophil chemotaxis, such as IL6 or CXCL2; however, future studies are required to confirm this.

Recent studies have shown that in response to a variety of stimuli, neutrophils become primed or activated leading to an increase in surface expression of adhesion receptors including LFA-1 (CD11a/CD18), Mac-1 (CD11b/CD18), and L-selectin (CD62L) [77,78,79,80]. CD66a has been speculated to play a key role in the activation of adhesion receptors and has shown to be involved in neutrophil recruitment, activation, and adhesion [35,36,37]. This increased surface expression promotes chemotaxis and adhesion and, therefore, is critical for the neutrophils to reach the site of inflammation [77,81].

We therefore assessed activation markers in neutrophils in the bone marrow, circulation and kidney tissue following IRI. This analysis of neutrophil cell surface marker was performed as previously described [81,82,83]. Accordingly, activated neutrophil were defined by having increased expression of cell surface markers including CD66a and CD11b. Neutrophil surface expression of CD66a increased in the bone marrow and kidney tissue and CD11b increased in kidney tissue confirming neutrophil activation and increased adhesion marker expression following IRI. BRD4 inhibition suppressed up-regulation of CD66a in the kidney tissue suggesting a reduction in neutrophil activation. No increase of either marker was noted in the blood, suggesting that the activated neutrophils transmigrated into the kidney tissue. Similarly, in experimental studies of peritonitis, decrease of neutrophils expressing CD66a and CD11b in blood coincided with an increase at the site of inflammation [81].

While it has previously been thought that neutrophils undergo apoptosis at the site of inflammation and are phagocytosed by macrophages, it is now understood that neutrophils undergo a process termed reverse migration, and re-enter the vasculature [84,85]. Previous studies have shown that neutrophils reverse migrate primarily to two sites: the lung and the bone marrow [72,86,87]. In this regard, neutrophils expressing CD66a increased in the bone marrow at 24 and 48 h post-IRI when compared to control mice suggesting potential reverse migration as has been visualized by Kubes et al. [72].

Interestingly, BRD4 inhibition suppressed neutrophil up-regulation of CD66a in response to IRI. CD66 has been shown to activate β2-integrins and play a role in neutrophil recruitment and arrest, while treatment with anti-CD66a monoclonal antibodies significantly decreased neutrophil adhesion to stimulated HUVEC [35,37]. Consistent with our results, a previous study demonstrated NFκB regulation of CD66a expression [88]. These results, combined with our in vitro adhesion studies, suggest that MS417 targets CD66a on the neutrophils, leading to decreased surface expression of adhesion markers, preventing neutrophil adhesion and transmigration to the site of injury.

Our final observation was that BRD4 inhibition reduces the severity of kidney injury following IRI, and this injury correlated with neutrophil infiltration. BRD4 inhibition reduced the expression of both kidney injury markers, HAVCR1 and LCN2, and decreased injury to the tubules. While HAVCR1 and NGAL are not directly mediated by NFκB, previous studies targeting NFκB signaling have demonstrated a subsequent down-regulation in expression of these genes [89,90,91]. JQ1 treatment prior to IRI has also attenuated tubular epithelial cell injury, shown by reduced both swelling and loss of brush borders [34]. We speculate that the protective effect of BRD4 inhibition on IRI was mediated, at least in part, by reducing neutrophil accumulation, and therefore neutrophil mediated tissue damage.

Our study had some key strengths. Our study confirmed previous observations, albeit in different systems, around BRD4 inhibition affecting neutrophil adhesion and infiltration [55,57,76]. In addition, we extended the understanding of neutrophil biology to show BRD4 inhibition not only has an effect on neutrophil infiltration at the site of injury, but also in the bone marrow and in the circulation. Finally, we demonstrated BRD4 inhibition affects neutrophil activation as demonstrated by FACS analysis. Taken together, these observations strongly set the stage for embarking on translational studies in the clinical setting, especially due to the safety profile of these drugs, particularly those used in cancer treatment.

Our study has some important limitations. We did not relate the impact of early treatment on longer term kidney outcomes. Interestingly, we observed that MS417 inhibits TGF-β-induced gene expression, suggesting that kidney fibrosis following IRI may also be attenuated by MS417 treatment. Furthermore, the current study focused on the relationship between inflammation injury and neutrophil accumulation. Other cell types, including monocytes, play a key role in inflammation following IRI; however, we did not assess the impact of BRD4 inhibition on these cell types [92,93]. From a mechanistic perspective, we can only infer that prevention of neutrophil adhesion by BRD4 inhibition is via blockade of CD66, but other mechanisms may also be important. From a translational perspective, our study is complicated by a design that looked at MS417 pre-treatment prior to injury. This was purposeful and designed to overcome the short half-life of BET inhibitors and achieve higher tissue concentrations prior to IR I [94]. This limits the translation of our work to clinical settings in which IRI is an expected complication allowing pre-treatment of the BRD4 inhibitor to be possible [95,96,97]. This has been described in cases of cisplatin-induced nephrotoxicity where pre-treatment protocols were able to reduce oxidative stress and inflammation [98,99,100] and in humans in the setting of aortic aneurysm repair [101].

In summary, we report that BRD4 inhibition with MS417 attenuates neutrophil adhesion to endothelial cells in vitro, and reduces neutrophil counts in the bone marrow and kidney, attenuates activated neutrophils in the kidney, and reduces tubular injury following IRI, in vivo. Taken together, these findings suggest that targeting NFκB by inhibiting BRD4 may be a potential therapeutic approach to IRI.

## 4. Materials and Methods

### 4.1. In Silico Analysis

NFκB-mediated genes identified by Pahl et al. were compared to genes differentially expressed in kidney tissue at 4, 24 and 48 h following bilateral IRI in mice identified by Liu et al. to discover 34 NFκB-mediated genes expressed following IRI [19,20]. Heatmap analysis was performed by converting the FPKM values into z-scores and performing Euclidean hierarchical clustering using the R package pheatmap. Log-transformed fold change and independent Student’s t-tests (*p* < 0.05) were performed by comparing the individual time points (4, 24 and 48 h) to sham mice. False discovery rate was corrected with Benjamini and Hochberg multiple testing correction (q < 0.05). Significantly enriched biological processes were identified using Biological Networks Gene Ontology with Benjamini and Hochberg multiple testing correction (*p* < 0.05) and then run on Enrichment Map with Jaccard coefficient of 0.5 (*p* = 0.001; false discovery rate *q*-value = 0.05).

### 4.2. Cell Culture

Immortalized proximal tubular HK-2 cells were cultured in equal parts Dulbecco’s Modified Eagle Medium (DMEM) and Ham’s F12 media supplemented with 10% fetal bovine serum, 10ng/mL epidermal growth factor, 5 μg/mL transferrin, 5 μg/mL insulin, 0.05 μM hydrocortisone, 50 units/mL penicillin, and 50 μg/mL streptomycin and incubated at 37 °C with 5% CO_2_ as previously described [102,103].

Human primary renal proximal tubular epithelial cells (PTECs) were purchased from Lonza, Basel, Switzerland. The cells were cultured in DMEM/F12 media supplemented with 10% fetal bovine serum, 10 ng/mL epidermal growth factor, 5 μg/mL transferrin, 5 μg/mL insulin, 0.05 μM hydrocortisone, 50 units/mL penicillin, and 50 μg/mL streptomycin and incubated at 37 °C with 5% CO_2_ as previously described [102,104]

Primary Umbilical Vein Endothelial Cells; Normal, Human, Pooled (HUVEC) (ATCC^®^ PCS100013™) were purchased from the American Type Culture Collection (Manassas, VA, USA). HUVEC were grown to 80% confluency in Endothelial Cell Growth Medium (PromoCell GmbH, Heidelberg, Germany) and maintained at 37 °C and at room air oxygen tension (21% O_2_) and 5% CO_2_ as previously described [105].

### 4.3. Luciferase Activity Assay

Cells were transfected with Renilla luciferase control reporter vector pRL-TK and a luciferase reporter for either NFκB, AP-1 or SBE (SMAD binding element) vector and incubated with fresh growth medium (DMEM/F-12) for 24 h, and then serum starved (serum-free DMEM/F-12) for 24 h. Cells were then treated with 10 ng/mL TNFα (NFκB), 50 ng/mL EGF (AP-1), or 5 ng/mL TGF-β (SBE) as a positive control for 6 h, 1 μM MS417 for 6 h, or a combination of 1 μM MS417 for 1 h then the varying positive control for 6 h. Control group was treated with serum free DMEM/F-12 for 6 h. Reporter activities were measured using the Promega, Madison Wisconsin dual-luciferase assay kit. The luciferase activity was normalized to the Renilla luciferase activity.

### 4.4. H_2_O_2_ Time Course

Human primary proximal tubular epithelial cells were cultured until passage 6, serum starved for 24 h and then treated with 200 μM H_2_O_2_ in DMEM/F12 for 0 (control), 1, 6, or 24 h. Media was removed; cells were washed twice with PBS and then collected using a cell scraper. Cells were spun down at 14,000 rpm for 2 min and the pellet was stored at −80 °C. In a subsequent set of experiments, cells were pre-treated with 1 μM MS417 for 2 h, then 200 μM H_2_O_2_ for 6 h.

### 4.5. Isolation of Primary Human Neutrophils

The protocol for human participation for research was reviewed and approved by The Hospital for Sick Children Research Ethics Board (approval code: 1000060065; approval date: 19-03-2019, Toronto, ON, Canada. The protocol followed the guidelines from the Canadian “Tri-Council Policy Statement (TCPS2): Ethical Conduct for Research Involving Humans (2018)”. Written, informed consent was obtained from all participants before participation and experiments were performed as previously described [31]. Neutrophils were isolated using Polymorphprep (Axis-shield, Oslo, Norway) following the manufacturers’ instructions. Equal volume of blood was added over Polymorphprep solution and centrifuged to allow gradient separation of the layers. The high density layer was collected and resuspended in 0.9% NaCl. Lysis of the red blood cells was performed by adding hypotonic (0.2% NaCl) and then hypertonic (1.6% NaCl) solutions for 1 min each. Prior to experiments, neutrophils were re-suspended in Hank’s Balanced Salt Solution (HBSS) with 1 mM CaCl_2_ and 1 mM MgCl_2_. All experiments were performed immediately after neutrophil isolation to minimize their in vitro activation.

### 4.6. Hypoxia/re-oxygenation of Endothelial Cells

Hypoxia/re-oxygenation was performed as previously described [31]. Hypoxic conditions were achieved by exposing primary HUVEC to 1% O_2_ and balanced N_2_ at 37 °C. PO_2_ within the hypoxic chamber was calibrated and monitored during experiments using a Proox 110 oxygen controller system (Biospherix, Parish, NY, USA). Cells were exposed to 2 h of hypoxia followed by treatment with 1 μM MS417 and varying periods of re-oxygenation ranging from 3 to 24 h, as indicated.

### 4.7. Neutrophil-endothelial Adhesion Assay

Neutrophil-endothelial adhesion assays were performed as previously described [31]. Freshly isolated human neutrophils were labelled with Calcein-AM (Thermo Fisher Scientific, Rockford, IL, USA) and incubated with HBSS alone or with 1 μM MS417 for 1 hr at 37 °C. Neutrophils (10^5^ cells/well) were then incubated with confluent HUVEC monolayers stimulated with TNFα and allowed to adhere for 30 min. Following incubation, non-adherent cells were removed by centrifuging the plate upside down. Adhesion was measured using a fluorescent plate reader at excitation and emission wavelengths of 494 and 517 nm, respectively. In another set of experiments, either neutrophils or HUVEC alone were treated with MS417 for 1 h, and unbound MS417 was removed prior to performing adhesion assays.

### 4.8. Cell Viability Assay

Cell viability was assessed using a MTT assay kit I (Roche Diagnostics purchased from Sigma-Aldrich, Oakville, ON, Canada). The calorimetric MTT Assay was performed on freshly isolated human neutrophils in a 96-well format (10^5^ neutrophils per well) as per manufacturers’ instructions. Neutrophils were treated with varying concentrations of MS417 ranging from 1 nM to 1 mM for 1 hr at 37 °C. At endpoint, MTT labeling reagent (a tetrazolium salt (3-(4,5-dimethylthiazol-2-yl)-2,5-diphenyltetrazolium bromide or MTT) was added to each well at a final concentration 0.5 mg/mL. Cells were incubated for 4 h at 37 °C before adding 100 μL per well of the solubilization solution and incubated over night at 37 °C. Cell viability was measured by reading the absorbance on a microplate reader at 600 nm. The reference wavelength of 660 nm was used for background reduction. Cell viability following hypoxia and varying periods of re-oxygenation with 1 μM MS417 treatment was also measured. HUVEC were exposed to 2 h of hypoxia followed by treatment with 1 μM MS417 and 0, 6, 12 and 24 h of re-oxygenation with cell viability being measured as previously explained.

### 4.9. Quantitative PCR

Total RNA was extracted using an RNeasy Mini kit (Qiagen, Mississauga, ON, Canada) following manufacturers’ instructions. Isolated RNA samples were reverse-transcribed using QuantiTect Reverse Transcription (Qiagen) following manufacturers’ instructions. Quantitative polymerase chain reaction was run on ViiA 7 Real-Time PCR System (Life Technologies, Burlington, ON, Canada) using Taqman Fast Advanced Master Mix (Life Technologies). Human primers (Life Technologies) were purchased for the following genes: CCL2 (MCP1), Hs00234140_m1; CXCl2, Hs00601975_m1; IL6, Hs00174131_m1; and 18s, Hs03003631_g1. Mouse primers (Life Technologies) were purchased for the following genes: CCL2 (MCP1), Mm0044124_m1; TNFα, Mm00443260_g1; HAVCR1 (KIM1), Mm00506686_m1; LCN2 (NGAL), Mm01324470_m1; ICAM1, Mm00516023_m1; VCAM1, Mm01320970_m1; and GAPDH, Mm99999915_g1. Reaction mixtures were subjected to the amplification procedure as per the manufacturers’ instructions.

### 4.10. Mouse Renal IRI

All animal experiments were approved by the University of Toronto Faculty of Medicine Animal Care Committee and the Research Ethics Board (protocol #20011495) and conducted in accordance to Canadian Council on Animal Care (CCAC) guidelines. C57BL/6 male mice (8–10 weeks of age; Charles River Laboratories, Wilmington, MA, USA) were randomized into groups and the groups were randomized on the day of surgery (*n* = 7–11/group). Mice were treated with 1 μM MS417 or saline (control) by oral gavage daily for 7 days prior to IRI. The core temperature of the mice was maintained between 34 °C and 36 °C with a heating pad, and a midline incision was made under anesthesia (isoflurane 2–3%). The renal pedicle was exposed and clamped for 45 min. The clamp was removed, allowing reperfusion of the kidneys for 6, 24, and 48 h. Mice were maintained at 30–32 °C until fully recovered. Sham-treated mice underwent the same procedure without clamping of the renal pedicle. At the time of sacrifice, mice were euthanized and kidneys, blood and bone marrow (femur) were collected.

### 4.11. Immunohistochemistry

The middle third of the kidney subjected to IRI was placed into 10% neutral buffered formalin (Sigma-Aldrich) for immunohistochemistry analysis. Fixed kidney tissue was then paraffin-embedded, sectioned, stained and scanned. Histopathological injury to the tubules was assessed by PAS-staining three-μm sections. Neutrophil infiltration was assessed by Ly6B.2 clone 7/4 antibody staining (Bio-Rad Laboratories Inc., Mississauga, ON, Canada). Stained sections were visualized using the OlympusIX81 microscope and analyzed with ImageJ software. All sections were analyzed and quantified in a blinded manner by an experienced pathologist. Tubular injury was assessed over the entire outer medulla (corticomedullary border) based on tubular necrosis or luminal slough. Scores were assigned from 0 to 4 according to the percentage of involvement: 0—none; 0.5—<5%; 1—5 to 25%; 2—25 to 50%; 3—50 to 75%; 4—75 to 100%; 0.5 is added for any inner medullary tubular injury, and 0.5 is added for any cortical tubular injury. Semi-quantitative of neutrophil infiltration was assessed over the entire outer medulla (corticomedullary border) based on the proportion of area occupied by neutrophils. Scores were assigned from 0 to 4 according to the percentage of involvement: 0—none; 0.5—<5%; 1—5 to 25%; 2—25 to 50%; 3—50 to 75%; 4—75 to 100%; 0.5 is added for any inner medullary infiltration, and 0.5 is added for any cortical infiltration.

### 4.12. Flow Cytometry

After IRI, flow cytometry was used to quantify the infiltrating neutrophils and their subsets in the bone marrow, blood and kidney tissue. Sample fixation, labeling and flow cytometric analysis of mouse samples were performed as previously described [81]. Kidneys were maintained in cold PBS, and cells were recovered following manual filtration through a 40 μm nylon filter (Thermo Fisher Scientific, Rockford, IL, USA) using the plunger from a 10 mL syringe. Whole blood, bone marrow lavage fluid and kidney tissue were fixed with 1.6% formaldehyde for 15 min on ice. RBCs were lysed with Pharm Lyse (BD Biosciences, San Jose, CA, USA), and the cells were re-suspended in fluorescence activated cell sorting (FACS) buffer (Hanks^−/−^, 1% bovine serum albumin, 2 mM EDTA). Cells were blocked with rat serum (60–80 μg; Sigma-Aldrich) and mouse immunoglobulin IgG (2 μg; Sigma-Aldrich) and then labeled with an antibody master mix consisting of specific Cluster of Differentiation (CD) markers (CD66a, CD11b, CD62L, CD5, CD55 and CD101). Sample acquisition was performed using an LSR Fortessa or X-20 (BD Biosciences, San Jose, CA, USA) flow cytometer. Minimum of 2 × 10^5^ gated events were acquired in each sample. Mouse neutrophils were gated as L Ly6G^+ve^/F4/80^−ve^. Negative staining for each antibody was established with appropriate fluorescence minus one (FMO) controls. Absolute neutrophils counts were determined as the product of total cell counts and the percentage gated neutrophils. Absolute neutrophil counts were per femur for BM, per 200 μL for blood, and per half kidney.

### 4.13. Statistical Analysis

All results are expressed as mean ± standard error of mean (SEM). One-way analysis of variance (ANOVA) followed by Tukey’s multiple comparisons test was used unless otherwise stated. All statistical analysis was performed with GraphPad Prism 7 Software (GraphPad Software, La Jolla, CA, USA). A *p*-value of less than 0.05 was considered significant with * *p* < 0.05, ** *p* < 0.01, *** *p* < 0.001, **** *p* < 0.0001.

## Figures and Tables

**Figure 1 ijms-21-09620-f001:**
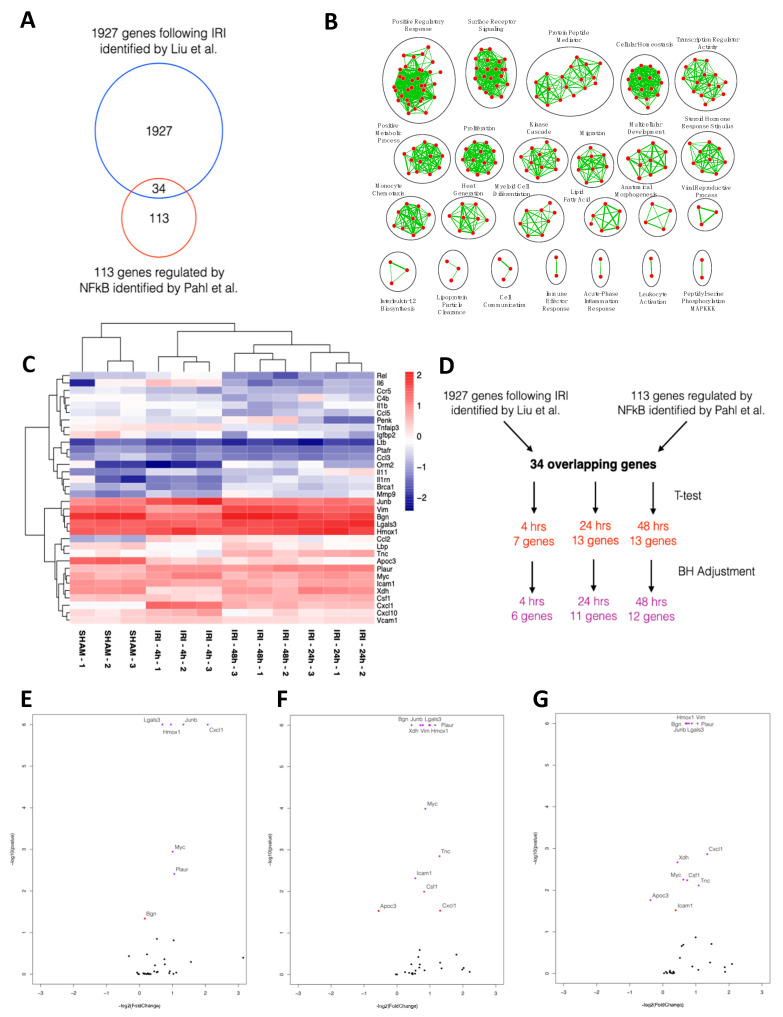
Early IRI is characterized by NFκB-mediated gene expression. (**A**) Venn diagram of genes identified following IRI and genes regulated by NFκB [19,20]. (**B**) Significantly enriched biological processes for the NFκB-mediated genes identified following IRI using Biological Networks Gene Ontology with Benjamini and Hochberg multiple testing correction (*p* < 0.05) and then run on Enrichment Map with Jaccard coefficient of 0.5 (*p* = 0.001; false discovery rate q-value = 0.05). Edge thickness represents the number of overlapping markers between two connecting nodes. (**C**) Heatmap representation of 34 NFκB-mediated genes following IRI at 4, 24 and 48 h with Euclidean hierarchical clustering. Log-transformed FPKM values for genes were converted into z-scores using mean values. Higher scores are indicated in red and lower scores in blue. (**D**) Flow diagram showing significant genes mediated by NFκB at 4, 24 and 48 h following IRI. (**E**–**G**) Volcano plots of NFκB-mediated genes following IRI at 4 (**E**), 24 (**F**), and 48 h (**G**). Independent Student t-tests (*p* < 0.05; red) with Benjamini and Hochberg multiple testing correction (*q* < 0.05; purple) was performed at each timepoint.

**Figure 2 ijms-21-09620-f002:**
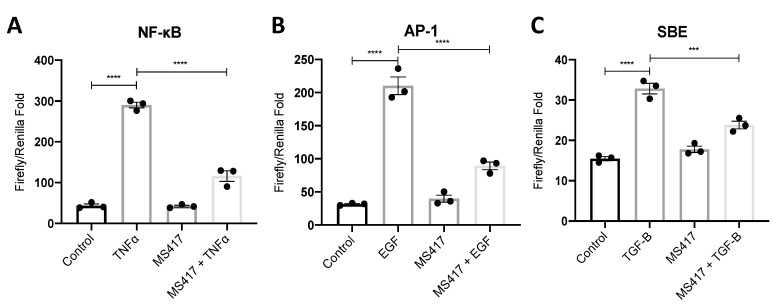
BRD4 inhibition attenuated NFκB, AP-1, and Smad-Binding Element (SBE) mediated luciferase activity. HK-2 cells were transfected with NFκB (**A**), AP-1 (**B**) or SBE (**C**) vectors and treated with 1 μM MS417 for 1 h followed by 10 ng/mL TNFa (NFκB), 50 ng/mL EGF (AP-1), or 5 ng/mL TGF-β1 (SBE) treatment. *n* = 3, *** *p* < 0.001, **** *p* < 0.0001.

**Figure 3 ijms-21-09620-f003:**
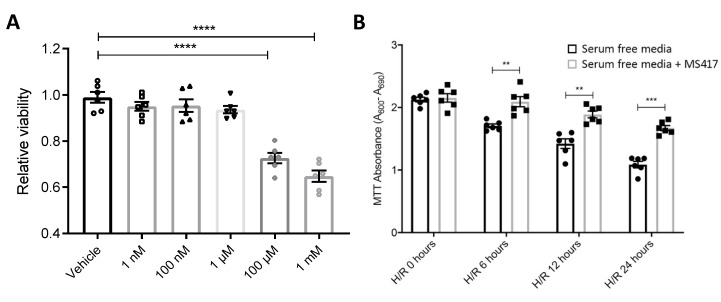
MS417 treatment had no effect on cell viability up to 1 μM (**A**) BRD4 inhibition with MS417 had no effect on neutrophil cell viability at concentrations up to 1 μM MS417. MTT Cell Viability assay was performed following treatment of isolated human neutrophils with 1nM–1 mM MS417 for 1 h. (**B**) BRD4 inhibition improves cell viability of HUVEC following H/R. HUVEC were exposed to 1% O_2_ for 2 h followed by treatment with 1 μM MS417 and 6, 12, and 24 h of re-oxygenation. Serum free media is in black, serum free media plus MS417 is in grey. Two-way ANOVA with Tukey was performed. *n* = 6, ** *p* < 0.01, *** *p* < 0.001, **** *p* < 0.0001.

**Figure 4 ijms-21-09620-f004:**
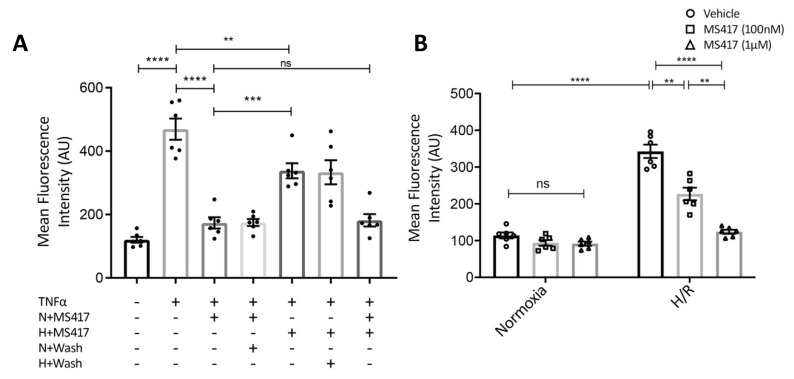
BRD4 inhibition attenuated neutrophil adhesion to the endothelium (**A**) BRD4 inhibition attenuated neutrophil adhesion to the endothelium with treatment to the neutrophils, endothelial cells, or combined. Isolated human neutrophils (10^5^ cells/well) or TNFα-stimulated HUVEC monolayers were treated with 1 μM MS417 for 1 h or left untreated and then co-incubated in various combinations as indicated. In some cases, MS417 was washed out as indicated. Cells were allowed to adhere for 30 min and any non-adherent cells were removed. Adhesion was measured using a fluorescent plate reader at excitation and emission wavelengths of 494 and 517 m, respectively. (N = neutrophils, H = HUVEC). (**B**) To better represent IRI, HUVEC were exposed to 1% O_2_ for 2 h followed by 3 h of re-oxygenation. Neutrophil were treated with either 100 nm or 1 μM of MS417 and adhesion was measured as previously described. Two-way ANOVA with Tukey was performed. *n* = 6, ** *p* < 0.01, *** *p* < 0.001, **** *p* < 0.0001.

**Figure 5 ijms-21-09620-f005:**
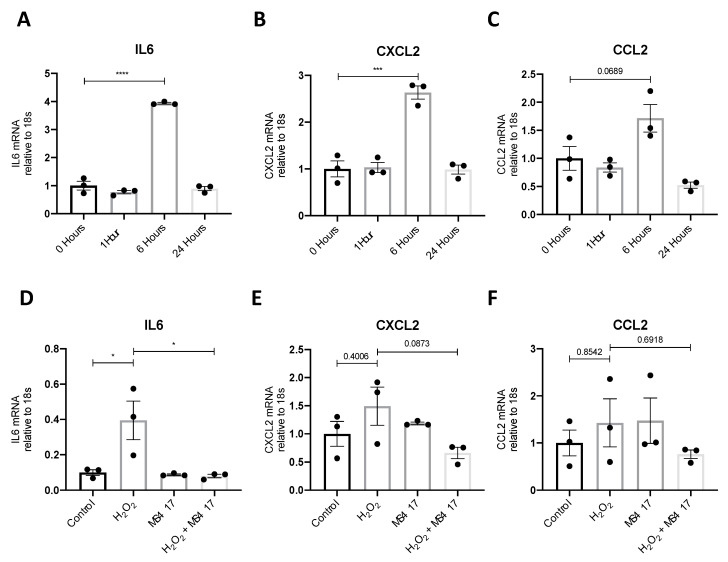
BRD4 attenuated the gene expression of NFκB-mediated genes. (**A**–**C**) Exposure time was optimized by treating PTECs with H_2_O_2_ for 0, 1, 6, and 24 h before cells were collected to qPCR analysis of IL6 (**A**), CXCL2 (**B**) and CCL2 (**C**). (**D**–**F**) In a subsequent set of studies, cells were treated with 1 μM MS417 for 2 h followed by H_2_O_2_ for 6 h followed by qPCR. *n* = 3, * *p* < 0.05, *** *p* < 0.001, **** *p* < 0.0001.

**Figure 6 ijms-21-09620-f006:**
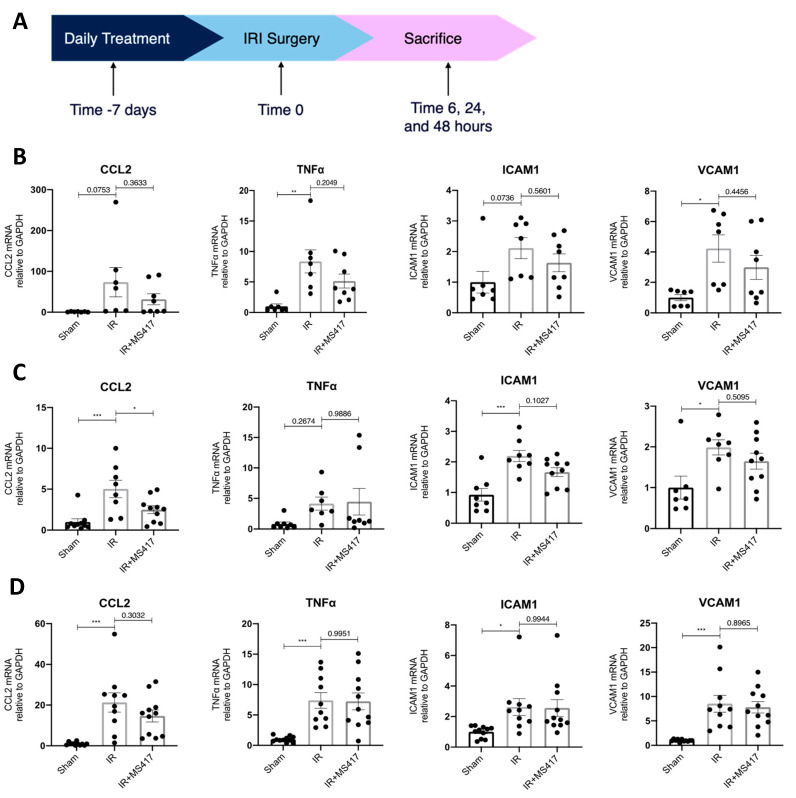
BRD4 inhibition attenuated gene expression in vivo following IRI. (**A**) A schematic representing the in vivo experimental design. C57BL/6 mice were treated with 1 μM MS417 daily for 7 days by oral gavage before unilateral IRI. Mice were sacrificed at 6 (**B**), 24 (**C**), and 48 (**D**) hours post-IRI and kidney tissue was collected for qPCR analysis of pro-inflammatory cytokines (CCL2 and TNFα) and adhesion molecules (ICAM1 and VCAM1). *n* = 7–11, * *p* < 0.05, ** *p* < 0.01, *** *p* < 0.001.

**Figure 7 ijms-21-09620-f007:**
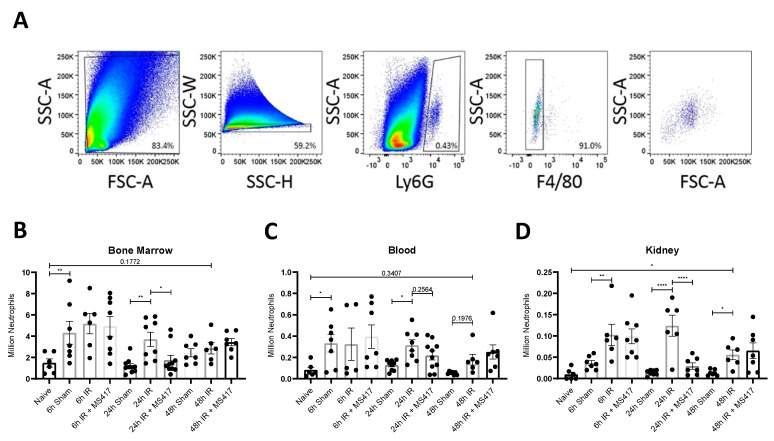
BRD4 inhibition attenuated absolute neutrophil counts in the bone marrow, circulation and kidney tissue following IRI (**A**) Flow cytometry gating strategy in kidney tissue. Mouse PMNs were gated based on Ly6G^+ve^/F4/80^−ve^. Doublets were excluded using SSC-W x SSC-H. (**B**–**D**) C57BL6 mice were treated with 1 μM MS417 daily for 7 days by oral gavage before unilateral IRI. Bone marrow (**B**), blood (**C**) and kidney tissue (**D**) was collected at 6, 24, and 48 h following IRI and absolute neutrophil counts were quantified by flow cytometry. ANOVA with Fischers LSD test was performed. *n* = 7–11, * *p* < 0.05, ** *p* < 0.01, **** *p* < 0.0001.

**Figure 8 ijms-21-09620-f008:**
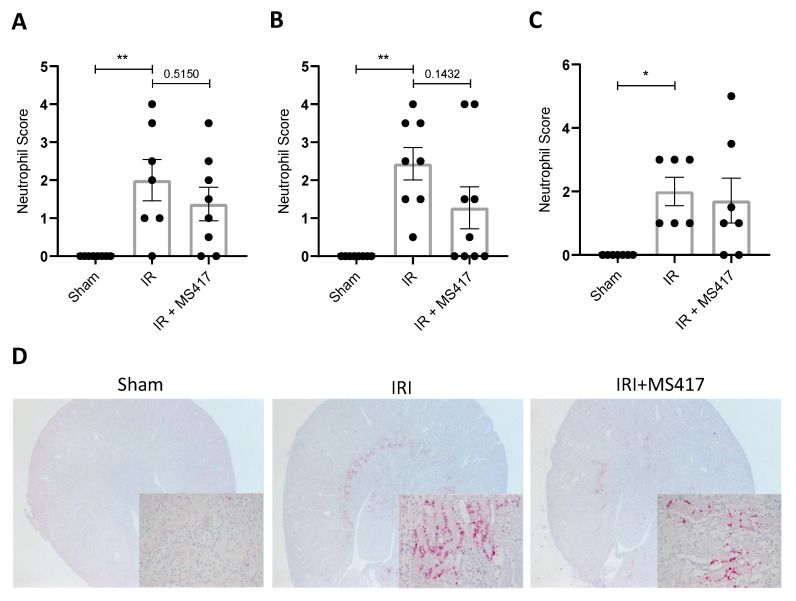
BRD4 inhibition attenuated neutrophil infiltration in kidney tissue following IRI (**A**–**C**) Semi-quantitative scoring of neutrophil infiltration on Ly6B.2 clone 7/4 stained kidney sections from 6 h (**A**), 24 h (**B**), and 48 h (**C**) performed in a blinded manner. (**D**) Representative images of neutrophil infiltration on Ly6B.2 clone 7/4 stained kidney sections performed at 24 h in Sham, IRI and IRI + MS417 mice. Images are at 1.6× magnification with high magnification inserts at 20×. *n* = 7–9, * *p* < 0.05, ** *p* < 0.01.

**Figure 9 ijms-21-09620-f009:**
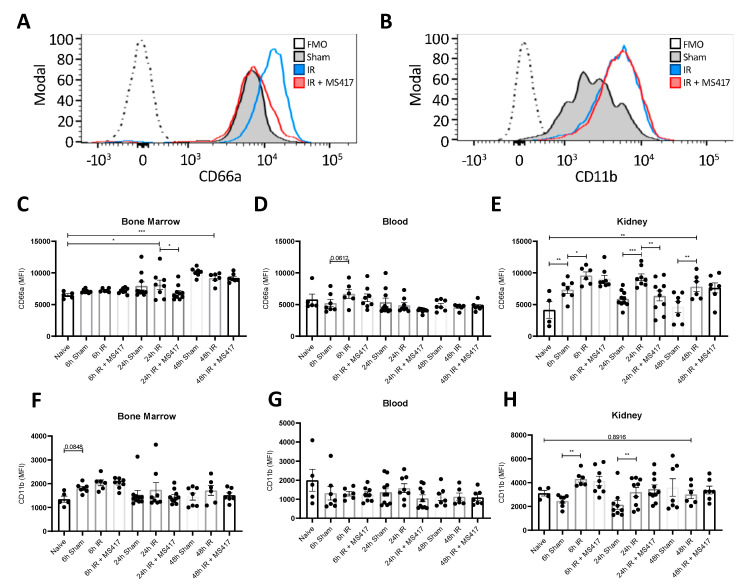
BRD4 inhibition attenuated CD66a up-regulation on kidney neutrophils following IRI but had no effect on CD11b expression. C57BL/6 mice were treated with 1 μM MS417 daily for 7 days by oral gavage before unilateral IRI. (**A**,**B**) Representative histograms of CD66a and CD11b expression for kidney tissue neutrophils (CD16^high^/side scatter area (SSC-A)^high^) in mice comparing Sham and IRI plus/minus MS417 treatment at 24 h post-IRI. Bone marrow (**C**,**F**), blood (**D**,**G**) and kidney tissue (**E**,**H**) was collected at 6, 24, and 48 h following IRI and CD66a+ and CD11b+ neutrophils were quantified by flow cytometry. MFI = mean fluorescence intensity. ANOVA with Fischers LSD test was performed. *n* = 7–11, * *p* < 0.05, ** *p* < 0.01, *** *p* < 0.001.

**Figure 10 ijms-21-09620-f010:**
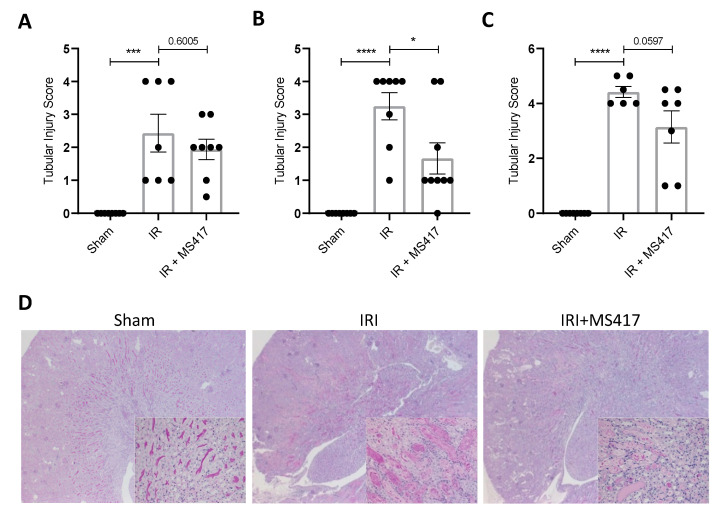
BRD4 inhibition attenuated tubular injury following IRI. (**A**–**C**) Semi-quantitative scoring of tubular injury on PAS sections performed in a blinded manner at 6 h (**A**), 24 h (**B**), and 48 h (**C**). (**D**) Representative images of PAS staining performed at 24 h in Sham, IRI and IRI + MS417 mice. Images are at 1.6× magnification with high magnification inserts at 20×. *n* = 7–9, * *p* < 0.05, *** *p* < 0.001, **** *p* < 0.0001.

**Figure 11 ijms-21-09620-f011:**
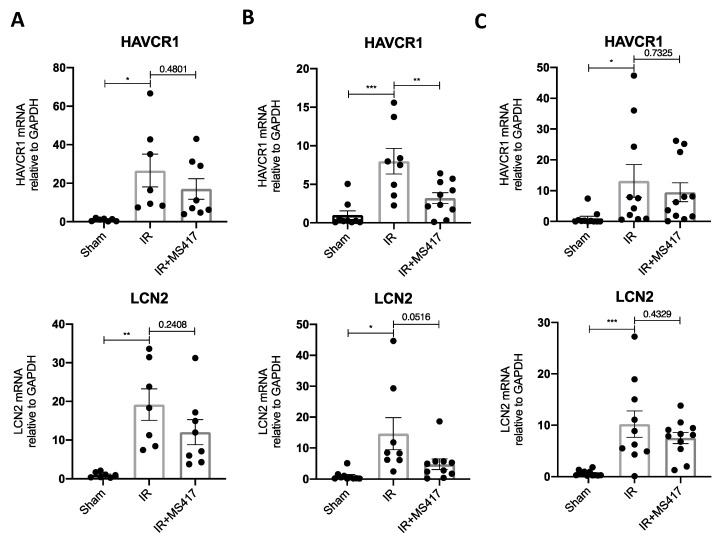
BRD4 inhibition attenuated expression of injury markers following IRI. (**A**–**C**) Mice were sacrificed at 6 (**A**), 24 (**B**), and 48 h (**C**) and kidney tissue was collected for qPCR analysis of injury markers (HAVCR1 and LCN2). *n* = 7–11, * *p* < 0.05, ** *p* < 0.01, *** *p* < 0.001.

**Figure 12 ijms-21-09620-f012:**
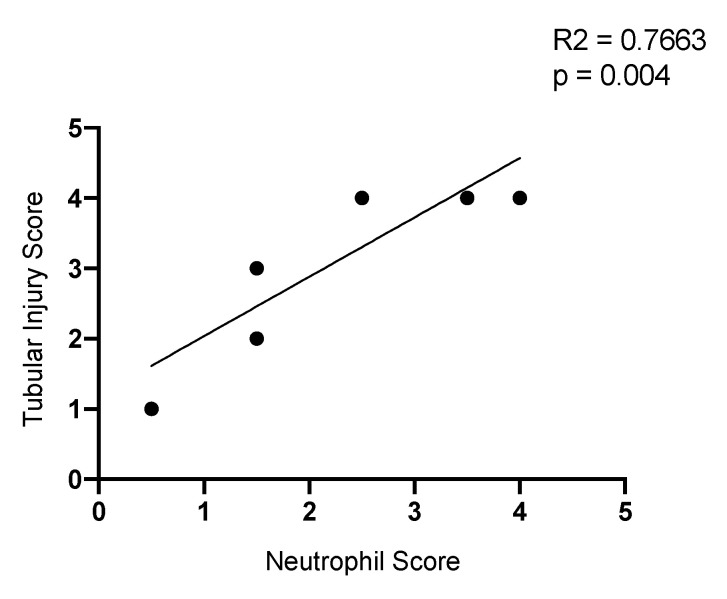
Neutrophil infiltration is related to tubular injury following IRI. Correlation plot of semi-quantitative scoring performed in a blinded manner of tubular injury and neutrophil infiltration at 24 h-post IR (R2 = 0.7663, *p* = 0.004). *n* = 8 (two sets of mice had the same scores so are overlapped).

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
