# Peer review of "Inhibition of BRD4 Reduces Neutrophil Activation and Adhesion to the Vascular Endothelium Following Ischemia Reperfusion Injury"

_ijms, 2020, doi:10.3390/ijms21249620_

Round 1

Reviewer 1 Report

In this work, Reid et al. investigated the role of BRD4 and its link with neutrophils in the process of renal ischemia-reperfusion injury (IRI). This is a well-written manuscript with a good structure and interesting results, pointing to an important role of BRD4, NFkB and neutrophils in renal IRI. I understand that the work benefits from a specific focus on NFkB and neutrophils, but the findings presented are likely caused by additional effects as well. In my opinion, this should be better addressed in the discussion:

  • I personally like that the authors used a BRD4 inhibitor, leading to NFkB attenuation rather than an direct NFkB inhibition. However, this implies that the effects by the inhibitor could also be dependent on additional downstream effects of MS417/BRD4 inhibition. As it is pointed out in Fig. 2, MS417 also affects AP-1 and SBE signaling (and there are likely other targets as well). As such, please discuss the results in the context of the broader spectrum of BRD4 inhibition.
  • In parallel, the manuscript focuses on neutrophil accumulation and activation. However, NFkB-mediated cytokine production will also influence the recruitment of other immune cells (e.g. macrophages) and the observed effects are thus likely not only because of altered neutrophil activation. Again, I understand that their is this specific focus in the research presented, but this broader view should be better addressed in the discussion.

Please also better define "early" renal IRI (as already mentioned in the abstract). What can be considered "early" renal IRI. The in vivo experiments actually show an effect mostly at 24h post-reperfusion, and not at 6h. In this model of 45 min of clamping followed by reperfusion, can 24h still be considered "early". How would you define "6h"? In addition, do the results imply that NFkB is not important in the very acute injury happening at 6h post-reperfusion? 

In addition, please also define the samples of which the results were used for the in silico analysis. What type of samples? What time points? Why can they be considered "early" gene expression changes?  

Please remove the statement that "some reduction" of CCL2 was observed in Fig. 5f. This is not at all significant. This figure could actually benefit from one or more additional experiments to reach a better statistical conclusion. 

It is unclear why the authors chose to use HK2 cells in Fig. 2 and PTECs in Fig. 5. Please explain. 

Please provide also insets in the histological images of Sham an MS417-treated kidneys in Fig. 8d and 10d, so to better compare the differences.

Regaring the effects on KIM1 and NGAL, is this known to be dependent on NFkB?

Reviewer 2 Report

The authors show that BRD4 inhibition with MS417 attenuates NFκβ mediated neutrophil activation and adhesion to endothelial cells in ischemic conditions. In the discussion, the authors have already mentioned and commented on the major limitations of the study, which improves the esteem of the manuscript. Nonetheless, I would like to add the following minor remarks and suggestions:

  1. The authors should mention in the introduction some background on MS417 and discuss why this inhibitor was chosen (and not JQ1 for example).
  2. Why is the neutrophil count shown in absolute numbers and not percentages? And how was total cell count achieved (tryhan blue staining). It would be more convenient to get absolute neutrophil numbers with the use of counting beads.
  3. Figure 1 e-f: I assume the x-axis designation is wrong and should be changed to log2(FoldChange) as this is the most convenient and easiest to interpret way to show differential expression data.
  4. Figure 4a: The x-as is not readable and should be fixed. Moreover, the authors should consider to change the denotation to N+MS417 instead of N-MS417 to improve compressibility.
  5. Figure 4b: Data from treatment with 100nM MS417 are shown although this is not mentioned in the rest of the paper.
  6. Materials and methods, immunohistochemistry: The authors should complete the following information: Based on which criteria tubular injury was scored? Which Ly6G+ antibody was used? Clarify how semi-quantitative scoring of neutrophils was done.

Author Response

Reviewer #2

The authors show that BRD4 inhibition with MS417 attenuates NFκβ mediated neutrophil activation and adhesion to endothelial cells in ischemic conditions. In the discussion, the authors have already mentioned and commented on the major limitations of the study, which improves the esteem of the manuscript. Nonetheless, I would like to add the following minor remarks and suggestions:

  1. The authors should mention in the introduction some background on MS417 and discuss why this inhibitor was chosen (and not JQ1 for example).

Response: In accord with this comment, we have revised the manuscript to provide more background on MS417 (Line 61).

We chose MS417 because it was designed to specifically inhibit BRD4 from interacting and binding to the lysine-310 residue on the acetylated RelA, effectively inhibiting transcription from occurring. Other BET inhibitors, such as JQ1, bind generically to acetyl-lysine residues on nuclear proteins leading to the possible inhibition of other pathways (Filippakopoulos et al., 2010). In addition, MS417 has been shown to limit chronic kidney injury in murine models of HIV nephropathy and diabetic nephropathy through reduced expression of NFκB target genes and improved renal function (Zhang et al., 2012; Liu et al., 2014).

  1. Why is the neutrophil count shown in absolute numbers and not percentages? And how was total cell count achieved (tryhan blue staining). It would be more convenient to get absolute neutrophil numbers with the use of counting beads.

Response:  The absolute neutrophil counts were quantified based on flow cytometry. Cells were counted during flow cytometry based on the gating strategy of LY6G+/F480-. This analysis yielded an absolute count.

  1. Figure 1 e-f: I assume the x-axis designation is wrong and should be changed to log2(FoldChange) as this is the most convenient and easiest to interpret way to show differential expression data.

Response:  We are grateful for the reviewers comments about Figure 1e-f. We revised the manuscript and have changed the x-axis. We agree that this is the best way to illustrate differential expression data.

  1. Figure 4a: The x-as is not readable and should be fixed. Moreover, the authors should consider to change the denotation to N+MS417 instead of N-MS417 to improve compressibility.

Response: Unfortunately, the files we submitted were compressed, and this interfered with the ability to read the axis in Figure 4a. We have revised Figure 4a in order to make the x-axis more readable. In addition, we changed the axis labelling to N+MS417/H+MS417 as suggested by the reviewer.

  1. Figure 4b: Data from treatment with 100nM MS417 are shown although this is not mentioned in the rest of the paper.

Response: We only performed a dose-response of MS417 in experiments that examined neutrophil adhesion to endothelial cell monolayers following hypoxia/reoxygenation. We have revised the text to emphasize the dose of MS417 used in each of the individual experiments.

  1. Materials and methods, immunohistochemistry: The authors should complete the following information: Based on which criteria tubular injury was scored? Which Ly6G+ antibody was used? Clarify how semi-quantitative scoring of neutrophils was done.

Response:

Tubular injury was assessed over the entire outer medulla (corticomedullary border) based on tubular necrosis or luminal slough.  Scores were assigned from 0 to 4 according to the percentage of involvement: 0 - none; 0.5 - <5%; 1 - 5 to 25%; 2 - 25 to 50%; 3 - 50 to 75%; 4 - 75 to 100%; 0.5 is added for any inner medullary tubular injury, and 0.5 is added for any cortical tubular injury.

The manuscript has been revised to include these detailed descriptions of the methodology used to perform the semi-quantitative scoring (Line 613).

Semi-quantitative of neutrophil infiltration was assessed over the entire outer medulla (corticomedullary border) based on the proportion of area occupied by neutrophils.  Scores were assigned from 0 to 4 according to the percentage of involvement: 0 - none; 0.5 - <5%; 1 - 5 to 25%; 2 - 25 to 50%; 3 - 50 to 75%; 4 - 75 to 100%; 0.5 is added for any inner medullary infiltration, and 0.5 is added for any cortical infiltration.

The manuscript has been revised to include these detailed descriptions of the methodology used to perform the semi-quantitative scoring (Line 617).

In the immunohistochemistry studies, we stained the kidney sections with the following antibody: Ly6B.2, clone 7/4 (Bio-Rad Laboratories, Inc.; MCA771GT). The manuscript has been revised to specify the antibody used for these studies (Line 610).

References:

Filippakopoulos, P., Qi, J., Picaud, S., Shen, Y., Smith, W. B., Fedorov, O., Morse, E. M., Keates, T., Hickman, T. T. and Felletar, I. (2010) 'Selective inhibition of BET bromodomains', Nature, 468(7327), pp. 1067.

Liu, R., Zhong, Y., Li, X., Chen, H., Jim, B., Zhou, M.-M., Chuang, P. Y. and He, J. C. (2014) 'Role of transcription factor acetylation in diabetic kidney disease', Diabetes, 63(7), pp. 2440-2453.

Zhang, G., Liu, R., Zhong, Y., Plotnikov, A. N., Zhang, W., Zeng, L., Rusinova, E., Gerona-Nevarro, G., Moshkina, N. and Joshua, J. (2012) 'Down-regulation of NF-κB transcriptional activity in HIV-associated kidney disease by BRD4 inhibition', Journal of biological chemistry, 287(34), pp. 28840-28851.